# Characteristics of Extracellular Vesicles and Preclinical Testing Considerations Prior to Clinical Applications

**DOI:** 10.3390/biomedicines10040869

**Published:** 2022-04-07

**Authors:** Min Heui Yoo, A-Ram Lee, Kyoung-Sik Moon

**Affiliations:** Department of Innovative Toxicology Research, Korea Institute of Toxicology, 141 Gajeon-ro, Yuseong-gu, Daejeon 34114, Korea; aram.lee@legochembio.com (A.-R.L.); ksmoon@kitox.re.kr (K.-S.M.)

**Keywords:** immunogenicity, tumorigenicity, soft agar colony formation assay, biodistribution, drug delivery agents, microRNAs, cytokines

## Abstract

Cell therapy products have significant limitations, such as storage instability, difficulties with transportation, and toxicity issues such as tumorigenicity and immunogenicity. Extracellular vesicles (EVs) secreted from cells show potential for therapeutic agent development. EVs have not been widely examined as investigational drugs, and non-clinical studies for the clinical approval of EV therapeutic agents are challenging. EVs contain various materials, such as DNA, cellular RNA, cytokines, chemokines, and microRNAs, but do not proliferate or divide like cells, thus avoiding safety concerns related to tumorigenicity. However, the constituents of EVs may induce the proliferation of normal cells; therefore, the suitability of vesicles should be verified through non-clinical safety evaluations. In this review, the findings of non-clinical studies on EVs are summarized. We describe non-clinical toxicity studies of EVs, which should be useful for researchers who aim to develop these vesicles into therapeutic agents. A new method for evaluating the immunotoxicity and tumorigenicity of EVs should also be developed.

## 1. Introduction

Extracellular vesicles (EVs) are heterogeneous small membrane structures that originate from plasma membranes. Although most EVs have a diameter of 50–200 nm, larger ones are also observed. Generally, particles up to a diameter of 1000 nm are regarded as EVs [1,2]. They are typically isolated from the conditioned media of cultured cells. The contents of EVs include proteins, mRNA, microRNA (miRNA), and nucleic acids [3]. Each vesicle performs a specific function in transferring biological material(s) to induce biological processes, such as replication, growth, apoptosis, and necrosis [4,5,6]. They are also required for cell-to-cell communication to maintain a normal homeostatic state [7]. EVs can be used as cargo carriers in physiological or pathological conditions and are considered biomarkers representing altered normal physiological states [8]. Based on these characteristics, EVs can be used for diverse purposes, from cosmetic to therapeutic applications. The main advantage of EVs is their limited adverse effects when used for therapeutic or cosmetic purposes, because they are composed of cell-derived materials and because of their potential for targeted cell delivery [9]. In addition, compared to cells, they are easier to store and transport.

The first EV that was identified is involved in transferrin receptor elimination, which plays a role in the maturation of reticulocytes, as reported by Harding et al. in 1983 [10]. The authors demonstrated the release of multi-vascular endosomes from the plasma membrane by exocytosis in rat reticulocytes. EVs can be found in all types of body fluids, such as plasma [11], bile [12], breast milk [13], urine [14], ascites, and cerebrospinal fluid [15]. Thus, these vesicles show potential for revealing abnormal conditions in various organs. EVs from the blood can be used to detect inflammation or an aberrant immune system, whereas those from breast milk can be utilized to diagnose breast conditions [16]. Halvaei et al. reported that EVs can be used for the diagnosis of various cancers using cancer-specific miRNAs [17]. Therefore, the types of EVs with clinical potential, the cells from which they are derived and relevant preclinical studies are described below and summarized in Figure 1.

## 2. Categories and Characteristics of EVs

### 2.1. Mesenchymal Stem/Stromal Cell-Derived EVs

Mesenchymal stem cells (MSCs) have been widely investigated as therapeutic options for various diseases, including graft versus host disease [18] and cardiac [19], neurological [20], and orthopedic [21] disorders. MSCs mainly reduce inflammation, enhance progenitor cell proliferation, improve tissue repair, and decrease infection. According to the U.S. Food and Drug Administration, over 35,000 clinical trials have been conducted in the USA, France, and Canada on cell-based therapies [22]. However, despite the potency of MSCs, numerous side effects, such as tumorigenesis and immunogenicity, have been reported in preclinical and clinical trials [23]. In addition, there are some limitations to the generation and storage of MSCs intended for use as therapeutics [24]. To maintain the efficacy of MSCs and overcome these drawbacks, MSC-derived EVs have received attention as therapeutic agents that can be used for renal protection and to manage various disorders, including cardiac dysfunction, myocardial infarction, stroke, hepatic fibrosis, and vascular proliferative diseases [25,26,27,28,29,30]. In particular, MSC-derived EVs are composed of factors such as cytokines, growth factors, RNA, and miRNAs, which originate from MSCs and thus exert similar effects to those of MSCs [31].

The effects of MSC-derived EVs in cancer cell biology are controversial [32]. Many groups have reported that MSC-derived EVs increase cancer proliferation, invasion, and metastasis. Bone marrow MSC-derived EVs were reported to stimulate the hedgehog signaling pathway in the growth of osteosarcoma and gastric cancer [33], whereas adipocyte MSC-derived EVs promoted breast cancer cell growth via activation of the Hippo signaling pathway [34]. However, adipose MSC-derived EVs inhibited prostate cancer growth by delivering miR-145 [35].

MSC-derived EVs of different origins show different effects in various diseases with divergent mechanisms. Further information is provided in Table 1.

### 2.2. Cancer-Derived EVs

EVs from tumor cells can be produced and utilized to stimulate or inhibit tumor growth under various conditions, depending on whether they will or will not be used for cancer treatment. Cancer-derived EVs can be detected in all bodily fluids, such as the blood, saliva, urine, and bile [14,46,47]. Based on this characteristic, many scientists have attempted to develop cancer-derived EVs as noninvasive biomarkers for diagnosing cancer in early stages of disease [48]. Specifically, cancer-derived EVs contain various biomarkers, such as miR-17, miR-19a, miR-21, miR-126/miR-141, miR-146, and miR-409, which have a range of effects on tumor growth and can be used for cancer diagnosis and prognosis [49,50,51,52].

The extracellular matrix, cancer-associated fibroblasts, inflammatory immune cells, and tumor-associated vasculature are components of the tumor microenvironment, which can be a major source of tumor-derived EVs [53,54]. Cancer-associated fibroblasts are among the major sources of tumor EVs with different effects before and after chemotherapy [55]. In particular, following chemotherapies, EVs derived from cancer-associated fibroblasts were shown to promote the chemoresistance and proliferation of colorectal and breast cancers [56,57]. EVs from tumors under hypoxic conditions enhanced angiogenesis and metastasis by modulating the microenvironment [58]. Because tumor-derived EVs contain important components, including nucleic acids and oncogenic proteins, they can be used as biomarkers for diagnosis, prognosis, therapeutic response prediction, and targeted therapy [4].

### 2.3. EVs as Anticancer Drug Delivery Agents

Jang et al. reported that EV-delivered doxorubicin had a greater effect on reducing tumor size than administration of pure doxorubicin in a colon adenocarcinoma xenograft model [59]. Furthermore, the use of an α_v_ integrin-specific iRGD peptide with EVs to deliver doxorubicin showed promising anticancer effects in an α_v_ integrin-positive breast cancer model [60]. Following the investigation of paclitaxel using an EV delivery system in a tumor xenograft model, Kim et al. reported its anticancer effects in vitro and in vivo [61]. Another group reported that EV-encapsulated paclitaxel directly targeted cancer stem cells that exhibited anticancer drug resistance [62]. EVs loaded with the antitumor drugs withaferin A or celastrol were administered to a human lung cancer xenograft mouse model, in which they showed anticancer effects [63,64]. Engineered EVs with superparamagnetic-conjugated transferrin have been shown to target tumor cells and reduce tumor growth in vivo [65]. In addition, an engineered anti-epidermal growth factor receptor nanobody fused with the EV anchor signal peptide glycosylphosphatidylinositol showed direct activity against tumor cells positive for epidermal growth factor receptor-positive tumor cells [66].

Because of their stability in biological fluids, EVs can escape from lung clearance and cross the blood-brain barrier [67,68], thus easily reaching tumors in various organs such as the liver, brain, and breast. Based on these characteristics, EVs can be used for cancer-targeting therapies.

## 3. Toxicity and Safety Assessment of EVs

Below, we review toxicity studies involving diverse organ-derived EVs according to the type of examination, namely, general toxicity, immunogenicity, tumorigenesis, and biodistribution tests. In addition, safety evaluations related to EVs are summarized in Table 2.

### 3.1. General Toxicity Tests

According to several reports related to general toxicology following the administration of EVs, rare general toxicity was observed in rodent and non-rodent testing. Bagno et al. analyzed the hematology index of rats that were intravenously injected with MSC-derived EVs and reported no adverse effects [19]. Welton et al. also reported the absence of abnormal clinical signs, abnormal body weight changes, abnormal changes in blood chemistry, and lesions on/in the tissues of mice after intravenous or intraperitoneal injection with HEK293T-derived EVs [15]. Sun et al. evaluated the safety of EVs derived from human umbilical cord MSCs (hucMSCs) using rats [69]. They intravenously infused hucMSCs into rats after inducing acute myocardial infarction to test the safety and efficacy of the EVs. The body weights and blood chemistry of the rats were analyzed to detect liver and kidney functions. They reported that the hucMSCs protected against weight loss from acute myocardial infarction and had no adverse effects on hepatic or renal function. Furthermore, Mendt et al. reported the safety of MSC-derived engineered EVs, which contained exogenously loaded siRNA, following long-term administration in mice [71]. The EVs (10^9^) were administered intraperitoneally every two days for four months into immunocompetent mice to evaluate potential toxicity. The researchers found no abnormalities following hematologic and chemical analyses of samples from the EV-treated groups compared with those from the vehicle control group. However, mild inflammation was observed in the liver, kidneys, lungs, brain, mesentery, and spleen of animals from both groups. In addition, minor toxicity was reported, with minimal to mild inflammation in the liver and kidneys after intravenous administration of BJ fibroblast (skin fibroblast)-derived EVs into C57BL/6 mice [23].

In summary, the efficacy of EVs is similar to that of the cells from which they were derived; however, there were fewer side effects for EVs than for the original cells [86]. Nonetheless, the preclinical toxicity test criteria for the filing of investigational new drug applications for these vesicles have not been clearly defined because of insufficient data. Whether conventional, general toxicity testing can detect the detailed toxicity due to EVs remains unclear. Therefore, detailed tools for evaluating the toxicity of EVs should be developed.

### 3.2. Immunogenicity/ Immunotoxicity Studies

Various preclinical studies have been conducted to evaluate safety based on the obvious efficacy of EVs, and rare notable immunogenicity has been reported [87,88,89]. Minor immune responses were reported by some researchers. For instance, Zhu et al. reported the immunogenicity of engineered HEK293T cell-derived EVs loaded with miRNA-199a-3p. To assess the EV-induced immune response, C57BL/6 mice were administered EVs intravenously and intraperitoneally for 3 weeks. At the end point, blood was harvested to examine hematology and immune markers, and spleen cells were collected to detect immunophenotypes. Minimal evidence of changes in immune markers was observed in the mice dosed with engineered EVs, but not with wild-type EVs [70]. Mendt et al. performed immunotyping of the spleen, bone marrow, and thymus in immunocompetent mice administered MSC-derived exosomes every 2 days via intraperitoneal injection for 3 weeks and found no significant changes in those mice compared to the non-treated mice [71]. Lu et al. reported that induced pluripotent stem cell (iPSC)-derived EVs induced less immunogenicity than iPSCs in non-human primates, such as rhesus macaques, which are similar to humans in terms of behavior, and immune system [90]. Moreover, because of the immunosuppressive characteristics of tumor-derived EVs, they were used for vaccination during tumor treatment. Specifically, dendritic cell vaccination using tumor-derived EVs has been shown to extend the survival time of WEHI3B-bearing mice [91]. Furthermore, tumor-derived EVs were shown to induce T-cell apoptosis, impairment of dendritic cell differentiation, and propagation of immunosuppressive myeloid suppressor cells, thus reducing natural killer cell activation [92,93]. Chalmin et al. showed that tumor-derived EVs had an immunosuppressive function in both mouse and human myeloid-derived suppressor cells to enhance the efficacy of cancer treatment [94].

In summary, the mechanism of action on the immune system differs depending on the origin and potential use of EVs. Therefore, it is necessary to clearly understand the characteristics of these vesicles on the immune system and perform toxicity tests. Additionally, EVs from most cell types contain major histocompatibility complex (MHC) molecules which are involved in antigen presentation [95]. EVs are not a single entity but contain multiple components that can induce immunogenicity or toxicity and interact with each other. In addition, because EVs are human-derived biopharmaceuticals, immunotoxicity/immunogenicity tests on animals as applied in small molecule investigations are not suitable. Therefore, to evaluate the immunogenicity and immunotoxicity of various cell-derived EVs, a powerful evaluation tool is necessary to predict their immunogenicity in the human body. We propose an evaluation method using human peripheral blood mononuclear cells (PBMCs) as a powerful tool to evaluate the immunogenicity of EVs.

### 3.3. Tumorigenicity Tests

Lu et al. studied iPSC-derived EVs rather than iPSCs because iPSCs can form teratomas after transplantation [90]. Their results showed that EVs posed no risk of teratoma formation even though their effects were similar to those of iPSCs in rhesus macaque monkeys following topical administration of a bolus dose of 50 μg EVs onto inflicted wounds. The results were evaluated using wound area analysis and histology after 14 days to detect the efficacy and formation of teratomas. Lee et al. revealed that EVs released by ectopic expression of EIF3C in human hepatocellular carcinoma promoted angiogenesis and tumorigenesis using a Huh7 xenograft model and a human umbilical vein endothelial cell tube formation model as in vivo and in vitro tests, respectively [96]. Vallabhaneni et al. reported that EVs derived from human MSCs accelerated tumor growth and metastasis with changes in the tumor microenvironment [97]. They used the MCF-7 xenograft model to test the effects of human MSC-derived EVs on the growth of breast tumors in an immunodeficient mouse model for 40 days. Larger tumors with increased angiogenesis were observed in the group administered MCF-7 cells together with EVs compared to those in mice administered MCF cells alone. In addition, larger tumors were observed in the group treated with MCF-7 cells and EVs than in mice treated with only MCF-4 cells, with increased angiogenesis as the main mechanism.

The effect of EVs on tumorigenesis varies depending on the state of the vesicle in terms of assembly, the source of cells from which the vesicles originate, and their contents. In addition, because the characteristics of EVs derived from different cell types show different effects in various cells, including cancer cells, existing techniques used in tumorigenicity tests of cell therapy products are not sufficient for evaluating EV-derived tumorigenicity. Therefore, an in vitro tumorigenicity test should be performed before in vivo tumorigenicity testing of EVs. The soft agar colony formation assay is an in vitro tumorigenicity assay that is useful for determining the effects of EVs on the tumorigenesis of tumor cells and normal cells (Figure 2). This test can be used to evaluate the growth of tumor cells and tumorigenesis of normal cells in an in vitro system.

### 3.4. Biodistribution Tests

Biodistribution analysis is an important safety evaluation method for determining the residual amount, residual position, and clearance time of biopharmaceuticals such as cell, gene, and EV therapies [98,99,100]. To evaluate safety during biodistribution assessments of gene and cell therapies, globally established preclinical studies are performed [74]. Periodic evaluation of the biodistribution of EVs is necessary for preclinical drug development, such as for gene and cell therapies. However, the biodistribution testing of EVs is challenging because of the complexity of detection methods and lack of precedent. Nevertheless, various methods for detecting the distribution of EVs have been suggested. Wiklander et al. determined the in vivo biodistribution of EVs based on the cell source, administration route, and targeting [101]. To evaluate the biodistribution of EVs, the authors used enhanced green fluorescent protein-positive EVs and DiR-labeled EVs. Over 80% of the intravenously injected EVs were detected in the liver for over 48 h. However, different delivery routes such as intraperitoneal (i.p.) and subcutaneous (s.c.) administration influenced the distribution pattern, and EVs were observed in both the liver and gastrointestinal tract following the abovementioned routes of administration. In contrast, after intravenous injection, EVs were not detected in the liver, which is a different pattern from i.p and s.c. administrations. Smyth et al. demonstrated the biodistribution and delivery efficiency of different cancer-cell-derived EVs [102]. Fluorescently labeled and radiolabeled EVs were administered to nude mice to analyze the biodistribution of vesicles in vivo. PC3- and MCR-7-derived EVs showed similar distributions to major organs such as the liver, spleen, and kidneys. EV levels in the blood disappeared within 24 h after systemic exposure, including following intravenous administration.

As such, the biodistribution of EVs depends on the route of administration and the cells from which they were derived. Therefore, it is essential to verify their safety by clearly describing the in vivo distribution and clearance time through preclinical studies. Currently, the above methods are used, and new methods are being developed.

## 4. Conclusions and Future Perspectives

We have described the features of and preclinical studies performed on EVs. These vesicles show advantages similar to those of the cells of origin but exhibit lower toxicity, such as reduced immunogenicity and tumorigenicity. Thus, several groups worldwide have embarked on developing EVs as therapeutic agents and have performed preclinical testing and clinical trials of these vesicles. EVs can withstand mass processing, quality control, storage, and management more easily than cells. Thus, EVs show potential for clinical applications as chemical drugs.

However, as mentioned above, the toxicity of these vesicles may vary depending on the cell type and target disease; therefore, preclinical studies based on the characteristics of the specific vesicle system are needed. This review provides useful information for researchers performing preclinical studies of EVs. We hope to devise a more general and powerful assessment tool or model capable of detecting the general toxicity, immunogenicity/immunotoxicity, and tumorigenicity of various cell-derived EVs in the future.

## Figures and Tables

**Figure 1 biomedicines-10-00869-f001:**
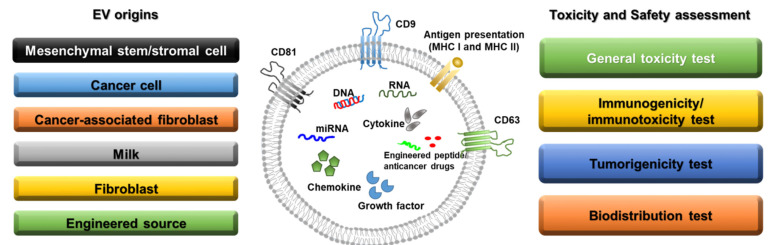
Sources of extracellular vesicles and toxicity/safety assessments. EVs can originate from mesenchymal stem/stromal cells, cancer cells, cancer-associated fibroblasts, milk, normal fibroblasts, and engineered cells. EVs have a lipid bilayer and can contain transmembrane proteins, antigen presentation proteins, DNA, RNA, miRNA, cytokines, chemokines, growth factors, engineered peptides, and anticancer drugs. Before clinical studies of EVs, general toxicity, immunogenicity, tumorigenicity, and biodistribution tests should be performed in preclinical studies depending on the source of the EVs.

**Figure 2 biomedicines-10-00869-f002:**
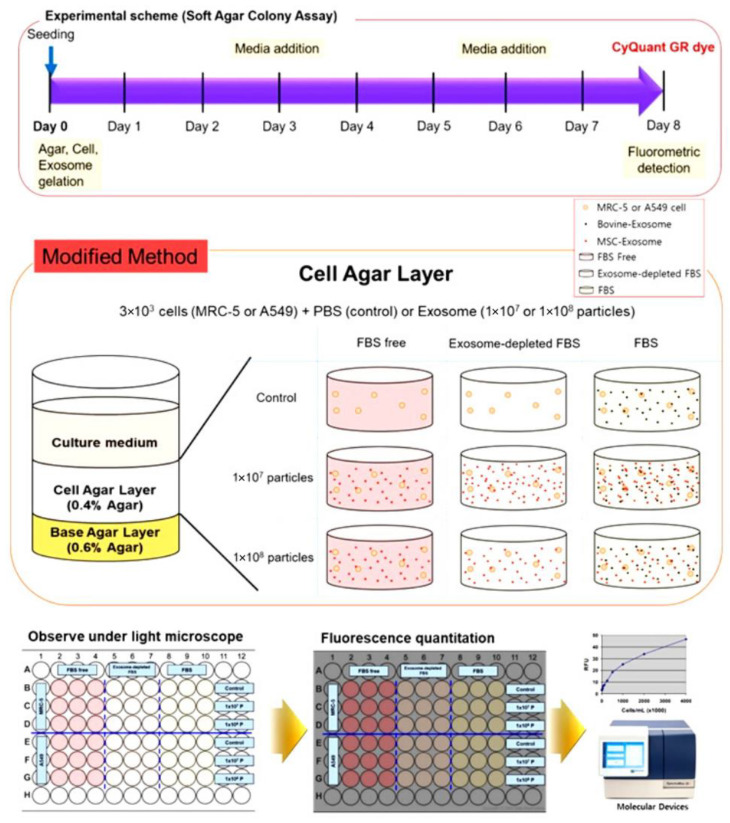
Experimental design of soft agar colony formation assay in vitro. For in vitro tumorigenicity test of EVs, MRC-5 and A549 cells (3 × 10^3^ cells/well) are divided into three groups (FBS Free, exosome-depleted FBS, and normal FBS contained well) and each group is assigned a PBS-treated control group. The EVs are incubated with cells on the cell agar layer as shown in the diagram, and the medium is replaced every 3 days. On the 8th day of culture, the agar is solubilized to dissolve both agar and cells, and after treatment with CyQuant GR dye, fluorescence is measured at 485/520 nm to determine whether colony formation increased. FBS, fetal bovine serum; PBS, phosphate-buffered saline.

**Table 1 biomedicines-10-00869-t001:** MSC-derived EVs of different origins with different effects in various diseases.

EV Origin	Target Disease	Mechanisms & Characteristics	Animals Used	Ref. No.
Bone marrow-derived mesenchymal stem cells	Wound healing	Promoting M2 polarization of macrophages miR-223 wound healing by transferring EV-derived microRNA	6–8 weeks old female C57BL/6 J mice	[28]
Mesenchymal stem cells	Alzheimer’s disease	Evaluating mouse cognitive deficits Stimulating neurogenesis in the subventricular zone Alleviating beta amyloid 1−42-induced cognitive impairment	7–8-week-old C57BL/6 mice	[36]
Adipose tissue-derived mesenchymal stem/stromal cells	Cisplatin-induced acute kidney injury	Protection of animals from death due to cisplatin-induced acute kidney injury	6-week-old male Sprague Dawley rats	[37]
Bone marrow-derived mesenchymal stem cells	Pilocarpine-induced status epilepticus	Neuroprotective and anti-inflammatory effects Increasing normal hippocampal neurogenesis and cognitive and memory function	6–8-week-old C57BL/6 mice	[38]
Mesenchymal stromal cells	A newborn rat model of bronchopulmonary dysplasia (BPD) induced by 14 days of neonatal hyperoxia exposure (85% O_2_)	Protecting from apoptosis, inhibiting inflammation, and increasing angiogenesis Preventing the disruption of alveolar growth, increasing small blood vessel number, and inhibiting right heart hypertrophy at P14, P21, and P56	Newborn rats	[39]
Embryonic mesenchymal stem cells	Critical-sized osteochondral defects (1.5 mm diameter and 1.0 mm depth)	Complete restoration of cartilage and subchondral bone	8-week-old female Sprague Dawley rats	[40]
Umbilical cord mesenchymal stem cells	Perinatal brain injury (hypoxic-ischemic and inflammatory with lipopolysaccharide)	Inhibiting the production of pro-inflammatory molecules and preventing microgliosis in rats with perinatal brain injury Decreasing TNF-α and IL-1β expression in injured brains	2-day-old Wistar rat pups	[41]
Umbilical Cord mesenchymal stem cells	CCl_4_-induced liver injury	Suppressing the development of liver tumors Inhibiting oxidative stress in liver tumors Reducing oxidative stress and inhibiting apoptosis in liver fibrosis	4–5-week-old female BALB/c mice	[42]
Mesenchymal stromal cells	Cavernous nerve injury (CNI)	Enhancing smooth muscle content and neuronal nitric oxide synthase (nNOS) in the corpus cavernosum Improving erectile function after CNI Increasing penile nNOS expression and alleviating cell apoptosis	10-week-old male Sprague Dawley rats	[43]
Mesenchymal stem cells	Traumatic Brain Injury (TBI) with a 20 mm cylindrical impactor hemorrhaged over 12.5 min using a Masterflex pump	Lowering Neurological Severity Score (NSS) (*p* < 0.05) during the first five days post-injury Faster full neurological recovery	35–45 kg female Yorkshire swine	[44]
Mesenchymal stem cells	UV-irradiated skin	Attenuating UV-induced histological injury and inflammatory response in mouse skin Preventing cell proliferation and collagen deposition in UV-irradiated mouse skin Increasing antioxidant activity	newborn and adult Kunming mice	[45]

**Table 2 biomedicines-10-00869-t002:** EV toxicity and safety assessment.

Types	Study Design	Results	Ref.
General toxicity	Intravenous injection of MSC-derived exosome to rats: analyzing hematological indexes	No side effects on hematology indexes	[69]
	Intravenous/intraperitoneal injection of HEK293T-derived exosomes to C57BL/6 mice: Gross necropsy, histopathology, hematology analyses	No abnormal clinical signs, no abnormal body weight changes, no abnormal changes in blood chemistry, and no lesions found in tissues	[70]
	Intravenous injection of BJ fibroblast-derived exosomes to C57BL/6: Toxicology and necropsy analyses	Minimal to mild inflammation in liver and kidney, but mild immune activation of immune system	[71]
	Skin sensitization, photosensitization, eye and skin irritation, and acute oral toxicity with adipose stem cells (ASC)-derived exosomes in Sprague Dawley rats	No side effects and toxicity	[72]
	Intravenous injection of HEK Expi293F-derived exosomes to BALB/c mice: hematology analysis, pathological macroscopic analysis (brain, heart, lungs, liver, kidney, pancreas, spleen, skeletal muscle(hind leg), thymus, mesenteric lymph node, duodenum, caecum, tail vein)	No signs of toxicity and immune response	[73]
Immunogenicity/ immunotoxicity	Intravenous/intraperitoneal injection of HEK293T-derived EVs to C57BL/6 mice: Analyzing spleen immunophenotyping and rodent MAP	No signs of toxicity, minimal evidence of changes in immune markers	[70]
	Exposure of leukocytes to MSC-derived or bovine milk-derived EVs: Leukocyte population assay	Both MSC-EV and BM-EV increased leukocyte proliferation by 1.8 to 2.5-fold in the presence of phytohemagglutinin	[74]
	Testing MSC-derived or bovine milk-derived EVs with plasma, HL-60 phagocytic cells, or RAW264.7 cells: complement activation assay, phagocytosis assay, or nitric oxide test	No complement activation elicited by MSC-EVs, while BM-EVs elicited 5-fold increase; neither BM-EVs nor MSC-EVs induced phagocytosis; no nitrite level changes with both EV types	[75]
	Systemic anaphylaxis of MSC-derived exosomes using guinea pigs	No systemic anaphylaxis response in guinea pigs	[69]
	Testing HEK293T-derived EVs with THP-1, U937 human monocytic cells: apoptosis/ necrosis assay, microsphere phagocytosis assay	Homeostatic level of apoptosis/necrosis maintained after EV exposure; lower EV dosage facilitated phagocytosis while no effect observed with higher EV dosage	[76]
	Testing HEK Expi293F-derived exosomes with human whole blood: human whole blood assay	Minimal cytotoxicity and pro-inflammatory cytokine response	[77]
	Testing fetal liver MSC-exosomes with NK differentiated from PBMCs: proliferation, cytotoxicity, intracellular phospho-Smad2/3 assay	Impaired natural killer cell function	[78]
Gene toxicity	Exposure of lymphocytes to MSC-derived or bovine milk-derived EVs: alkaline comet assay	Neither MSC-EVs nor BM-EVs significantly increased comet tail length	[79]
	Exposure of CHO-K1 Chinese hamster ovarian cells to MSC-derived or bovine milk-derived EVs: micronucleus assay	No increase in micronucleus-positive cells	[80]
	Testing TMZ-resistant exosomes with GBM: Alkaline comet assay	Chemoresistance to temozolomide in glioblastoma	[81]
Tumorigenicity	Exposure of HGC-27 gastric cancer cells to MSC-derived exosomes: transwell migration, invasion, cell colony-forming, and soft agar assays	MSC-exosomes promoted migration and invasion of HGC-27 cells and while MSC-exosomes enhanced the colony formation of HGC-27 cells in serum-free conditions and cell sphere formation in soft agar	[81]
	Subcutaneous injection of colorectal cancer stem cell (CRCSC)-exosomes to BALB/c mice: in vivo gene targeting, tumorigenicity assay, colony formation assay	Tumorigenesis and immunosuppressive tumor microenvironment in colorectal cancer	[82]
	Intravenous injection of melanoma-exosomes to B16F1 xenografted C57BL/6N mice: tumorigenicity test	Tumor progression	[83]
	Subcutaneous injection of MDA-MB-231-exosomes to SKOV3 and CoC1 xenografted BALB/c nude mice: tumorigenicity test	Tumor progression	[84]
	Intraperitoneal injection of both MDA-MB-231-exosomes and MDA-MB-231 to NOD/SCID nude mice: peritoneal carcinomatosis assay	Tumor progression	[85]

## Data Availability

Not applicable.

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
