# Peer review of "Characteristics of Extracellular Vesicles and Preclinical Testing Considerations Prior to Clinical Applications"

_biomedicines, 2022, doi:10.3390/biomedicines10040869_

Round 1

Reviewer 1 Report

The manuscript describes the potential use of EVs in clinical practice and underlines their safety and toxicity
characteristics and concludes with the lack of appropriate methods to assess the immunotoxicity and
tumorigenicity of EVs. Although the concept is interesting and the manuscript offers a very complete
description of the subject, it should be improved by adding specific details.

1. The title should clearly and accurately reflect the results of the manuscript. The authors should
consider improving the title.

2. The possibility of clinical applications is extremely interesting for this reason the authors should
consider the classification of EVs and the need for accurate methods to discriminate the different
subpopulations of EVs and to verify their specific cargo; Two very important aspects in considering
their clinical application. Moreover, the difficulty in pinpointing the cells producing EVs and
identifying the recipient cells is another aspect.

3. The tables appear too long. The text should be more concise. Possibly consider adding details (e.g.
specify the cell source of EVs).

4. What is the relevance of Figure 2? This is little discussed or discussed at the end. Please consider
commenting on the figure in the introduction.

Minor comment

Authors should review the English, there are errors in the text:

vasculoproliferative - vascular proliferative

glycosyl phosphatidyl inositol glycosylphosphatidylinositol

chemistry analyses- chemical analysis

Reviewer 2 Report

Yoo and co-workers provide an interesting review article on safety features of EVs when they are considered for clinical purposes. Dealing with EVs is a relatively recent scientific branch. Because they are of interest for clinical application, it is important to study and discuss the safety aspects of EVs in therapy.  The article reads good and provides a brief preclinical overview for scientists with a clinical focus when considering to use EVs for their projects. 

Minor issues:  Not all statements are supported by the provided literature

Introduction: 2 nd line, although most EVs have a diameter range of 50 nm – 200 nm, larger ones are also observed. Generally, particles up to a diameter of 1000 nm are regarded as EVs.

2 nd but last line of introduction: The breast milk statement lacks a  reference

Categories:

Table 1 needs a better/more specific title and maybe better definition in the table. It should be more concrete. If only MSC EVs are shown, as suggested by the title of the chapter, then say so in the title of the table as well. If this table provides a more general overview, then it would be nice to organize the table better (subtitles for different EV types). As it stands, the table is a little confusing.

Toxicity:

Table 2: Does it make sense to distinguish the toxicity data between syn-, all- and xenogenic experiments. EVs contain functional molecules (MHC..), which might matter in toxicity tests. The authors should comment this issue in the text. -  P.9: (3.4. biodistribution) The authors need to clarify the conditions of the EV enrichment. Are intravenously injected EVs enriched in the liver or not. I was confused.   
